# PAC-Bayes bounds for stable algorithms with instance-dependent priors

**Omar Rivasplata**
UCL

**Emilio Parrado-Hernández**
University Carlos III of Madrid

**John Shawe-Taylor**
UCL

**Shiliang Sun**
East China Normal University

**Csaba Szepesvári**
DeepMind

## Abstract

PAC-Bayes bounds have been proposed to get risk estimates based on a training sample. In this paper the PAC-Bayes approach is combined with stability of the hypothesis learned by a Hilbert space valued algorithm. The PAC-Bayes setting is used with a Gaussian prior centered at the expected output. Thus a novelty of our paper is using priors defined in terms of the data-generating distribution. Our main result estimates the risk of the randomized algorithm in terms of the hypothesis stability coefficients. We also provide a new bound for the SVM classifier, which is compared to other known bounds experimentally. Ours appears to be the first uniform hypothesis stability-based bound that evaluates to non-trivial values.

## 1   Introduction

This paper combines two directions of research: stability of learning algorithms, and PAC-Bayes bounds for algorithms that randomize with a data-dependent distribution. The combination of these ideas enables the development of risk bounds that exploit stability of the learned hypothesis but are independent of the complexity of the hypothesis class. The PAC-Bayes framework (Shawe-Taylor and Williamson [1997], McAllester [1999a,b]) is used here with 'priors' defined in terms of the data-generating distribution, as introduced by Catoni [2007] and developed further e.g. by Lever et al. [2010], Parrado-Hernández et al. [2012] and Dziugaite and Roy [2018]. Specifically, our work derives PAC-Bayes bounds for hypothesis stable Hilbert space valued algorithms.

The analysis introduced by Bousquet and Elisseeff [2002], which followed and extended Lugosi and Pawlak [1994] and was further developed by Celisse and Guedj [2016], Abou-Moustafa and Szepesvári [2017] and Liu et al. [2017] among others, shows that stability of learning algorithms can be used to give bounds on the generalisation of the learned functions. Intuitively, this is because stable learning should ensure that slightly different training sets give similar solutions. In this paper stability is measured by the sensitivity coefficients (see our Definition 1) of the hypothesis learned by a Hilbert space valued algorithm. We provide an analysis leading to a PAC-Bayes bound for randomized classifiers under Gaussian randomization. As a by-product of the stability analysis we derive a concentration inequality for the learned hypothesis. Applying it to Support Vector Machines (Shawe-Taylor and Cristianini [2004], Steinwart and Christmann [2008]) we deduce a concentration bound for the SVM weight vector, and a PAC-Bayes performance bound for SVM with Gaussian randomization. Experimental results compare our new bound with other stability-based bounds, and with a more standard PAC-Bayes bound. We also experiment with their use in model selection.

## 2  Definitions and Main Result(s)

We consider a learning problem where the learner observes pairs $(X_i, Y_i)$ of patterns (inputs) $X_i$ from the space[1] $\mathcal{X}$ and labels $Y_i$ in the space $\mathcal{Y}$. A training set (or sample) is a finite sequence $S_n = ((X_1, Y_1), \ldots, (X_n, Y_n))$ of such observations. Each pair $(X_i, Y_i)$ is a random element of $\mathcal{X} \times \mathcal{Y}$ whose (joint) probability law is[2] $P \in M_1(\mathcal{X} \times \mathcal{Y})$. We think of $P$ as the underlying 'true' (but unknown) data-generating distribution. Examples are i.i.d. (independent and identically distributed) in the sense that the joint distribution of $S_n$ is the $n$-fold product measure $P^n = P \otimes \cdots \otimes P$.

A learning algorithm is a function $\mathsf{A} : \cup_n (\mathcal{X} \times \mathcal{Y})^n \to \mathcal{Y}^{\mathcal{X}}$ that maps training samples (of any size) to predictor functions. Given $S_n$, the algorithm produces a learned hypothesis $\mathsf{A}(S_n) : \mathcal{X} \to \mathcal{Y}$ that will be used to predict the label of unseen input patterns $X \in \mathcal{X}$. Typically $\mathcal{X} \subset \mathbb{R}^d$ and $\mathcal{Y} \subset \mathbb{R}$. For instance, $\mathcal{Y} = \{-1, 1\}$ for binary classification, and $\mathcal{Y} = \mathbb{R}$ for regression. A loss function $\ell : \mathbb{R} \times \mathcal{Y} \to [0, \infty)$ is used to assess the quality of hypotheses $h : \mathcal{X} \to \mathcal{Y}$. Say if a pair $(X, Y)$ is sampled, then $\ell(h(X), Y)$ quantifies the dissimilarity between the label $h(X)$ predicted by $h$, and the actual label $Y$. We may write $\ell_h(X, Y) = \ell(h(X), Y)$ to express the losses (of $h$) as function of the training examples. The (theoretical) *risk* of hypothesis $h$ under data-generating distribution $P$ is $R(h, P) = \langle \ell_h, P \rangle$.[3] It is also called the *error* of $h$ under $P$. The *empirical risk* of $h$ on a sample $S_n$ is $R(h, P_n) = \langle \ell_h, P_n \rangle$ where $P_n = \frac{1}{n} \sum_{i=1}^{n} \delta_{(X_i, Y_i)}$ is the empirical measure[4] on $\mathcal{X} \times \mathcal{Y}$ associated to the sample. Notice that the risk (empirical or theoretical) is tied to the choice of a loss function. For instance, consider binary classification with the 0-1 loss $\ell_{01}(y', y) = \mathbf{1}[y' \neq y]$, where $\mathbf{1}[\cdot]$ is an indicator function equal to 1 when the argument is true and equal to 0 when the argument is false. In this case the risk is $R_{01}(c, P) = P[c(X) \neq Y]$, i.e., the probability of misclassifying the random example $(X, Y) \sim P$ when using $c$; and the empirical risk is $R_{01}(c, P_n) = \frac{1}{n} \sum_{i=1}^{n} \mathbf{1}[c(X_i) \neq Y_i]$, i.e., the in-sample proportion of misclassified examples.

Our main theorem concerns Hilbert space valued algorithms, in the sense that the learned hypotheses live in a Hilbert space $\mathcal{H}$. In this case we may use the Hilbert space norm $\|w\|_{\mathcal{H}} = \sqrt{\langle w, w \rangle_{\mathcal{H}}}$ to measure the difference between the hypotheses learned from two slightly different samples.

To shorten the notation we will write $\mathcal{Z} = \mathcal{X} \times \mathcal{Y}$. A generic element of this space is $z = (x, y)$, the observed examples are $Z_i = (X_i, Y_i)$ and the sample of size $n$ is $S_n = (Z_1, \ldots, Z_n)$.

**Definition 1.** *Consider a learning algorithm* $\mathsf{A} : \cup_n \mathcal{Z}^n \to \mathcal{H}$ *where* $\mathcal{H}$ *is a separable Hilbert space. We define*[5] *the* hypothesis sensitivity coefficient *of* $\mathsf{A}$ *at sample size* $n$ *as follows:*

$$\beta_n = \sup_{i \in [n]} \sup_{z_i, z_i'} \|\mathsf{A}(z_{1:i-1}, z_i, z_{i+1:n}) - \mathsf{A}(z_{1:i-1}, z_i', z_{i+1:n})\|_{\mathcal{H}} \, .$$

This is close in spirit to what is called "uniform stability" in the literature, except that our definition concerns stability of the learned hypothesis itself (measured by a distance on the hypothesis space), while e.g. Bousquet and Elisseeff [2002] deal with stability of the loss functional. The latter could be called "loss stability" (in terms of "loss sensitivity coefficients") for the sake of informative names.

Writing $z_{1:n} \approx z_{1:n}'$ when these $n$-tuples differ at one entry (at most), an equivalent formulation to the above is $\beta_n = \sup_{z_{1:n} \approx z_{1:n}'} \|\mathsf{A}(z_{1:n}) - \mathsf{A}(z_{1:n}')\|_{\mathcal{H}}$. In particular, if two samples $S_n$ and $S_n'$ differ only on one example, then $\|\mathsf{A}(S_n) - \mathsf{A}(S_n')\|_{\mathcal{H}} \leq \beta_n$. Thus our definition implies stability with respect to replacing one example with an independent copy. Alternatively, one could define $\beta_n = \operatorname{ess\,sup}_{S_n \approx S_n'} \|\mathsf{A}(S_n) - \mathsf{A}(S_n')\|_{\mathcal{H}}$, which corresponds to the "uniform argument stability" of Liu et al. [2017]. We avoid the 'almost-sure' technicalities by defining our $\beta_n$'s as the maximal difference (in norm) with respect to all $n$-tuples $z_{1:n} \approx z_{1:n}'$. The extension to sensitivity when changing several examples is natural: $\|\mathsf{A}(z_{1:n}) - \mathsf{A}(z_{1:n}')\|_{\mathcal{H}} \leq \beta_n \sum_{i=1}^{n} \mathbf{1}[z_i \neq z_i']$. Note that $\beta_n$ is a Lipschitz factor with respect to the Hamming distance. The "total Lipschitz stability" of Kontorovich [2014] is a similar notion for stability of the loss functional. The "collective stability" of London et al. [2013] is not comparable to ours (different setting) despite the similar look.

We will consider randomized classifiers that operate as follows. Let $\mathcal{C}$ be the classifier space, and let $Q \in M_1(\mathcal{C})$ be a probability distribution over the classifiers. To make a prediction the randomized classifier picks $c \in \mathcal{C}$ according to $Q$ and predicts a label with the chosen $c$. Each prediction is made with a fresh $c$ draw. For simplicity we use the same label $Q$ for the probability distribution and for the corresponding randomized classifier. The risk measures $R(c, P)$ and $R(c, P_n)$ are extended to randomized classifiers: $R(Q, P) \equiv \int_{\mathcal{C}} R(c, P) \, dQ(c)$ is the *average theoretical risk* of $Q$, and $R(Q, P_n) \equiv \int_{\mathcal{C}} R(c, P_n) \, dQ(c)$ its *average empirical risk*. Given two distributions $Q_0, Q \in M_1(\mathcal{C})$, the Kullback-Leibler divergence (a.k.a. relative entropy) of $Q$ with respect to $Q_0$ is

$$KL(Q \| Q_0) = \int_{\mathcal{C}} \log\left(\frac{dQ}{dQ_0}\right) dQ \, .$$

Of course this makes sense when $Q$ is absolutely continuous with respect to $Q_0$, which ensures that the Radon-Nikodym derivative $dQ/dQ_0$ exists. For Bernoulli distributions with parameters $q$ and $q_0$ we write $\mathrm{kl}(q \| q_0) = q \log\left(\frac{q}{q_0}\right) + (1 - q) \log\left(\frac{1-q}{1-q_0}\right)$, and $\mathrm{kl}_+(q \| q_0) = \mathrm{kl}(q \| q_0) \mathbf{1}[q < q_0]$.

## 2.1 Main theorem: a PAC-Bayes bound for stable algorithms with Gaussian randomization

**Theorem 2.** *Let* $\mathsf{A}$ *be a Hilbert space valued algorithm. Suppose that (once trained) the algorithm will randomize according to Gaussian distributions* $Q = \mathcal{N}(\mathsf{A}(S_n), \sigma^2 I)$. *If* $\mathsf{A}$ *has hypothesis stability coefficient* $\beta_n$, *then for any randomization variance* $\sigma^2 > 0$, *for any* $\delta \in (0, 1)$, *with probability* $\geq 1 - 2\delta$ *we have*

$$\mathrm{kl}_+(R(Q, P_n) \| R(Q, P)) \leq \frac{\frac{n \beta_n^2}{2\sigma^2} \left(1 + \sqrt{\frac{1}{2} \log\left(\frac{1}{\delta}\right)}\right)^2 + \log\left(\frac{n+1}{\delta}\right)}{n} \, .$$

The proof, see Section 4 below, combines stability of the learned hypothesis (as in our Definition 1) and a PAC-Bayes theorem quoted there for reference. Note that the randomizing distribution $Q$ is data-dependent. Literature on the PAC-Bayes framework for learning linear classifiers includes Germain et al. [2015] and Parrado-Hernández et al. [2012] with references. Applications of this framework to neural networks are given, e.g., by London [2017], Dziugaite and Roy [2017], and Dziugaite and Roy [2018]. The latter combines PAC-Bayes and a substantially different stability notion: they use distributional stability for choosing a prior in a data-dependent manner.

## 2.2 Application: a PAC-Bayes bound for SVM with Gaussian randomization

For a Support Vector Machine (SVM) with feature map $\varphi : \mathcal{X} \to \mathcal{H}$ into a separable Hilbert space $\mathcal{H}$, we may identify[6] a linear classifier $c_w(\cdot) = \mathrm{sign}(\langle w, \varphi(\cdot) \rangle)$ with a vector $w \in \mathcal{H}$. With this identification we can regard an SVM as a Hilbert space[7] valued mapping that based on a training sample $S_n$ learns a weight vector $W_n = \mathrm{SVM}(S_n) \in \mathcal{H}$. In this context, stability of the SVM's solution then reduces to stability of the learned weight vector.

To be specific, let $\mathrm{SVM}_\lambda(S_n)$ be the SVM that regularizes the empirical risk over the sample $S_n$ by solving the following optimization problem:

$$\arg\min_w \left( \frac{\lambda}{2} \|w\|^2 + \frac{1}{n} \sum_{i=1}^n \ell(c_w(X_i), Y_i) \right) . \tag{svm}$$

Our stability coefficients in this case satisfy $\beta_n \leq \frac{2}{\lambda n}$ (Example 2 of Bousquet and Elisseeff [2002], adapted to our setting). Then a direct application of our Theorem 2 together with a concentration argument for the SVM weight vector (see our Corollary 9 below) gives the following:

**Corollary 3.** *Let* $W_n = \mathrm{SVM}_\lambda(S_n)$. *Suppose that (once trained) the algorithm will randomize according to Gaussian[8] distributions* $Q = \mathcal{N}(W_n, \sigma^2 I)$. *For any randomization variance* $\sigma^2 > 0$, *for any* $\delta \in (0, 1)$, *with probability* $\geq 1 - 2\delta$ *we have*

$$\mathrm{kl}_+(R(Q, P_n) \| R(Q, P)) \leq \frac{\frac{2}{\sigma^2 \lambda^2 n} \left(1 + \sqrt{\frac{1}{2} \log\left(\frac{1}{\delta}\right)}\right)^2 + \log\left(\frac{n+1}{\delta}\right)}{n} \, .$$

In closing this section we mention that our main theorem is general in that it may be specialized to any Hilbert space valued algorithm. This covers any regularized ERM algorithm [Liu et al., 2017]. We applied it to SVM's whose hypothesis sensitivity coefficients (as in our Definition 1) are known. It can be argued that neural networks (NN's) fall under this framework as well. Then an appealing future research direction, with deep learning in view, is to figure out the sensitivity coefficients of NN's trained by Stochastic Gradient Descent. Then our main theorem could be applied to provide non-vacuous bounds for the performance of NN's, which we believe is very much needed.

## 3 Comparison to other bounds

For reference we list several risk bounds (including ours). They are in the context of binary classification ($\mathcal{Y} = \{-1, +1\}$). For clarity, risks under the 0-1 loss are denoted by $R_{01}$ and risks with respect to the (clipped) hinge loss are denoted by $R_{\text{hi}}$. Bounds requiring a Lipschitz loss function do not apply to the 0-1 loss. However, the 0-1 loss is upper bounded by the hinge loss, allowing us to upper bound the risk with respect to the former in therms of the risk with respect to the latter. On the other hand, results requiring a bounded loss function do not apply to the regular hinge loss. In those cases the clipped hinge loss is used, which enjoys boundedness and Lipschitz continuity.

### 3.1 P@EW: Our new instance-dependent PAC-Bayes bound

Our Corollary 3, with $Q = \mathcal{N}(W_n, \sigma^2 I)$, a Gaussian centered at $W_n = \text{SVM}_\lambda(S_n)$ with randomization variance $\sigma^2$, gives the following risk bound which holds with probability $\geq 1 - 2\delta$:

$$\text{kl}_+(R_{01}(Q, P_n) \| R_{01}(Q, P)) \leq \frac{2}{\sigma^2 \lambda^2 n^2} \left( 1 + \sqrt{\frac{1}{2} \log\left(\frac{1}{\delta}\right)} \right)^2 + \frac{1}{n} \log\left(\frac{n+1}{\delta}\right).$$

As will be clear from the proof (see Section 4 below), this bound is obtained from the PAC-Bayes bound (Theorem 4) using a prior centered at the expected weight.

### 3.2 P@O: Prior at the origin PAC-Bayes bound

The PAC-Bayes bound Theorem 4 again with $Q = \mathcal{N}(W_n, \sigma^2 I)$, gives the following risk bound which holds with probability $\geq 1 - \delta$:

$$\forall \sigma^2, \quad \text{kl}_+(R_{01}(Q, P_n) \| R_{01}(Q, P)) \leq \frac{1}{2\sigma^2 n} \|W_n\|^2 + \frac{1}{n} \log\left(\frac{n+1}{\delta}\right).$$

### 3.3 Bound of Liu et al. [2017]

From Corollary 1 of Liu et al. [2017] (but with $\lambda$ as in formulation (svm)) we get the following risk bound which holds with probability $\geq 1 - 2\delta$:

$$R_{01}(W_n, P) \leq R_{\text{hi}}(W_n, P) \leq R_{\text{hi}}(W_n, P_n) + \frac{8}{\lambda n} \sqrt{2 \log\left(\frac{2}{\delta}\right)} + \sqrt{\frac{1}{2n} \log\left(\frac{1}{\delta}\right)}.$$

We use Corollary 1 of Liu et al. [2017] with $B = 1$, $L = 1$ and $M = 1$ (clipped hinge loss).

### 3.4 Bound of Bousquet and Elisseeff [2002]

From Example 2 of Bousquet and Elisseeff [2002] (but with $\lambda$ as in formulation (svm)) we get the following risk bound which holds with probability $\geq 1 - \delta$:

$$R_{01}(W_n, P) \leq R_{\text{hi}}(W_n, P) \leq R_{\text{hi}}(W_n, P_n) + \frac{2}{\lambda n} + \left( 1 + \frac{4}{\lambda} \right) \sqrt{\frac{1}{2n} \log\left(\frac{1}{\delta}\right)}.$$

We use Example 2 and Theorem 17 (based on Theorem 12) of Bousquet and Elisseeff [2002] with $\kappa = 1$ (normalized kernel) and $M = 1$ (clipped hinge loss).

In Appendix C below there is a list of different SVM formulations, and how to convert between them. We found it useful when implementing code for experiments.

There are obvious differences in the nature of these bounds: the last two (Liu et al. [2017] and Bousquet and Elisseeff [2002]) are risk bounds for the (un-randomized) classifiers, while the first two (P@EW, P@O) give an upper bound on the binary KL-divergence between the average risks (empirical to theoretical) of the randomized classifiers. Of course inverting the KL-divergence we get a bound for the average theoretical risk. Also, the first two bounds have an extra parameter, the randomization variance ($\sigma^2$), that can be optimized. Note that P@O bound is not based on stability, while the other three bounds are based on stability notions. Next let us comment on how these bounds compare quantitatively.

Our P@EW bound and the P@O bound are similar except for the first term on the right hand side. This term comes from the KL-divergence between the Gaussian distributions. Our P@EW bound's first term improves with larger values of $\lambda$, which in turn penalize the norm of the weight vector of the corresponding SVM, resulting in a small first term in P@O bound. Note that P@O bound is equivalent to the setting of $Q = \mathcal{N}(\mu W_n/\|W_n\|, \sigma^2 I)$, a Gaussian with center in the direction of $W_n$, at distance $\mu > 0$ from the origin (as discussed by Langford [2005] and implemented by Parrado-Hernández et al. [2012]).

The first term on the right hand side of our P@EW bound comes from the concentration of the weight (see our Corollary 9). Lemma 1 of Liu et al. [2017] implies a similar concentration inequality for the weight vector, but it is not hard to see that our concentration bound is slightly better.

Finally, in the experiments we compare our P@EW bound with Bousquet and Elisseeff [2002].

## 4 Proofs

As we said before, the proof of our Theorem 2 combines stability of the learned hypothesis (in the sense of our Definition 1) and a well-known PAC-Bayes bound, quoted next for reference:

**Theorem 4.** (PAC-Bayes bound) *Consider a learning algorithm* $\mathsf{A} : \cup_n (\mathcal{X} \times \mathcal{Y})^n \to \mathcal{C}$. *For any* $Q_0 \in M_1(\mathcal{C})$, *and for any* $\delta \in (0, 1)$, *with probability* $\geq 1 - \delta$ *we have*

$$\forall Q \in M_1(\mathcal{C}), \quad \mathrm{kl}_+(R(Q, P_n)\|R(Q, P)) \leq \frac{KL(Q\|Q_0) + \log(\frac{n+1}{\delta})}{n}.$$

*The probability is over the generation of the training sample* $S_n \sim P^n$.

This is borrowed from Langford [2005], originally Theorem 1 of Seeger [2002]. See also Theorem 2.1 of Germain et al. [2009]. To use the PAC-Bayes bound, we will use $Q_0 = \mathcal{N}(\mathbb{E}[\mathsf{A}(S_n)], \sigma^2 I)$ and $Q = \mathcal{N}(\mathsf{A}(S_n), \sigma^2 I)$, a Gaussian distribution centered at the expected output and a Gaussian (posterior) distribution centered at the random output $\mathsf{A}(S_n)$, both with covariance operator $\sigma^2 I$. The KL-divergence between those Gaussians scales with $\|\mathsf{A}(S_n) - \mathbb{E}[\mathsf{A}(S_n)]\|^2$. More precisely:

$$KL(Q\|Q_0) = \frac{1}{2\sigma^2}\|\mathsf{A}(S_n) - \mathbb{E}[\mathsf{A}(S_n)]\|^2.$$

Therefore, bounding $\|\mathsf{A}(S_n) - \mathbb{E}[\mathsf{A}(S_n)]\|$ will give (via the PAC-Bayes bound of Theorem 4 above) a corresponding bound on the binary divergence between the average empirical risk $R(Q, P_n)$ and the average theoretical risk $R(Q, P)$ of the randomized classifier $Q$. Hypothesis stability (in the form of our Definition 1) implies a concentration inequality for $\|\mathsf{A}(S_n) - \mathbb{E}[\mathsf{A}(S_n)]\|$. This is done in our Corollary 8 (see Section 4.3 below) and completes the circle of ideas to prove our main theorem. The proof of our concentration inequality is based on an extension of the bounded differences theorem of McDiarmid to vector-valued functions discussed next.

### 4.1 McDiarmid's inequality for real-valued functions of the sample

To shorten the notation let's present the training sample as $S_n = (Z_1, \ldots, Z_n)$ where each example $Z_i$ is a random variable taking values in the (measurable) space $\mathcal{Z}$. We quote a well-known theorem:

**Theorem 5.** (McDiarmid inequality) *Let* $Z_1, \ldots, Z_n$ *be independent* $\mathcal{Z}$-*valued random variables, and* $f : \mathcal{Z}^n \to \mathbb{R}$ *a real-valued function such that for each* $i$ *and for each list of 'complementary' arguments* $z_1, \ldots, z_{i-1}, z_{i+1}, \ldots, z_n$ *we have*

$$\sup_{z_i, z_i'} |f(z_{1:i-1}, z_i, z_{i+1:n}) - f(z_{1:i-1}, z_i', z_{i+1:n})| \leq c_i.$$

*Then for every* $\epsilon > 0$, $\Pr\{f(Z_{1:n}) - \mathbb{E}[f(Z_{1:n})] > \epsilon\} \leq \exp\left(\frac{-2\epsilon^2}{\sum_{i=1}^n c_i^2}\right)$.

McDiarmid's inequality applies to a *real-valued* function of independent random variables. Next we present an extension to *vector-valued* functions of independent random variables. The proof follows the steps of the proof of the classic result above, but we have not found this result in the literature, hence we include the details.

## 4.2 McDiarmid's inequality for vector-valued functions of the sample

Let $Z_1, \ldots, Z_n$ be independent $\mathcal{Z}$-valued random variables and $f : \mathcal{Z}^n \to \mathcal{H}$ a function into a separable Hilbert space. We will prove that bounded differences *in norm*[9] implies concentration of $f(Z_{1:n})$ around its mean *in norm*, i.e., that $\|f(Z_{1:n}) - \mathbb{E}f(Z_{1:n})\|$ is small with high probability.

Notice that McDiarmid's theorem can't be applied directly to $f(Z_{1:n}) - \mathbb{E}f(Z_{1:n})$ when $f$ is vector-valued. We will apply McDiarmid to the real-valued $\|f(Z_{1:n}) - \mathbb{E}f(Z_{1:n})\|$, which will give an upper bound for $\|f - \mathbb{E}f\|$ in terms of $\mathbb{E}\|f - \mathbb{E}f\|$. The next lemma upper bounds $\mathbb{E}\|f - \mathbb{E}f\|$ for a function $f$ with bounded differences in norm. Its proof is in Appendix A.

**Lemma 6.** *Let $Z_1, \ldots, Z_n$ be independent $\mathcal{Z}$-valued random variables, and $f : \mathcal{Z}^n \to \mathcal{H}$ a function into a Hilbert space $\mathcal{H}$ satisfying the bounded differences property: for each $i$ and for each list of 'complementary' arguments $z_1, \ldots, z_{i-1}, z_{i+1}, \ldots, z_n$ we have*

$$\sup_{z_i, z_i'} \|f(z_{1:i-1}, z_i, z_{i+1:n}) - f(z_{1:i-1}, z_i', z_{i+1:n})\| \leq c_i \,.$$

*Then $\mathbb{E}\|f(Z_{1:n}) - \mathbb{E}[f(Z_{1:n})]\| \leq \sqrt{\sum_{i=1}^{n} c_i^2}$.*

If the vector-valued function $f(z_{1:n})$ has bounded differences in norm (as in Lemma 6) and $C \in \mathbb{R}$ is any constant, then the real-valued function $\|f(z_{1:n}) - C\|$ has the bounded differences property (as in McDiarmid's theorem). In particular this is true for $\|f(z_{1:n}) - \mathbb{E}f(Z_{1:n})\|$ (notice that $\mathbb{E}f(Z_{1:n})$ is constant over replacing $Z_i$ by an independent copy $Z_i'$) so applying McDiarmid's inequality to it, combining with Lemma 6, we get the following theorem:

**Theorem 7.** *Under the assumptions of Lemma 6, for any $\delta \in (0,1)$, with probability $\geq 1 - \delta$ we have*

$$\|f(Z_{1:n}) - \mathbb{E}[f(Z_{1:n})]\| \leq \sqrt{\sum_{i=1}^{n} c_i^2} + \sqrt{\frac{\sum_{i=1}^{n} c_i^2}{2} \log\left(\frac{1}{\delta}\right)} \,.$$

Notice that the vector $c_{1:n} = (c_1, \ldots, c_n)$ of difference bounds appears in the above inequality only through its Euclidean norm $\|c_{1:n}\|_2 = \sqrt{\sum_{i=1}^{n} c_i^2}$.

## 4.3 Stability implies concentration

The hypothesis sensitivity coefficients give concentration of the learned hypothesis:

**Corollary 8.** *Let A be a Hilbert space valued algorithm. Suppose A has hypothesis sensitivity coefficient $\beta_n$ at sample size $n$. Then for any $\delta \in (0,1)$, with probability $\geq 1 - \delta$ we have*

$$\|\mathsf{A}(S_n) - \mathbb{E}[\mathsf{A}(S_n)]\| \leq \sqrt{n}\, \beta_n \left(1 + \sqrt{\frac{1}{2} \log\left(\frac{1}{\delta}\right)}\right) \,.$$

This is a consequence of Theorem 7 since $c_i \leq \beta_n$ for $i = 1, \ldots, n$, hence $\|c_{1:n}\| \leq \sqrt{n}\, \beta_n$.

Last (not least) we deduce concentration of the weight vector $W_n = \mathrm{SVM}_\lambda(S_n)$.

**Corollary 9.** *Let $W_n = \mathrm{SVM}_\lambda(S_n)$. Suppose that the kernel used by SVM is bounded by $B$. For any $\lambda > 0$, for any $\delta \in (0,1)$, with probability $\geq 1 - \delta$ we have*

$$\|W_n - \mathbb{E}[W_n]\| \leq \frac{2B}{\lambda\sqrt{n}} \left(1 + \sqrt{\frac{1}{2} \log\left(\frac{1}{\delta}\right)}\right) \,.$$

Under these conditions we have hypothesis sensitivity coefficients $\beta_n \leq \frac{2B}{\lambda n}$ (we follow Bousquet and Elisseeff [2002], Example 2 and Lemma 16, adapted to our setting). Then apply Corollary 8.

# 5 Gaussian distributions over the Hilbert space of classifiers?

This section aims to provide a rigorous explanation for Gaussian randomization in Hilbert spaces, which has been used here and in several previous machine learning works. For instance in the setting of SVM classifiers with feature map $\phi : \mathcal{X} \to \mathcal{H}$, the output is a weight vector that lives in the Hilbert space $\mathcal{H}$. With the Gaussian kernel in mind, we are facing an infinite-dimensional separable $\mathcal{H}$, which upon the choice of an orthonormal basis $\{e_1, e_2, \ldots\}$ can be identified with the space[10] $\ell_2 \subset \mathbb{R}^{\mathbb{N}}$ of square summable sequences of real numbers, via the isometric isomorphism $\mathcal{H} \to \ell_2$ that maps the vector $w = \sum_{i=1}^{\infty} w_i e_i \in \mathcal{H}$ to the sequence $(w_1, w_2, \ldots) \in \ell_2$. Thus without loss of generality we may regard the feature map as $\phi : \mathcal{X} \to \ell_2 \subset \mathbb{R}^{\mathbb{N}}$.

The PAC-Bayes approach applied to SVMs says that instead of committing to the weight vector $W_n = \mathrm{SVM}(S_n)$ we will randomize by choosing a fresh $W \in \mathcal{H}$ according to some probability distribution on $\mathcal{H}$ for each prediction. Suppose the randomized classifier is to be chosen according to a Gaussian distribution. Although it commonly appears in the literature, it is worth wondering just what is a Gaussian distribution over the space $\mathcal{H} = \ell_2$.

Two possibilities come to mind for the Gaussian random classifier $W$: (1) according to a Gaussian measure on $\ell_2$, say $W \sim \mathcal{N}(\mu, \Sigma)$ with mean $\mu \in \ell_2$ and covariance operator $\Sigma$ meeting the requirements (positive, trace-class) for this to be a Gaussian measure on $\ell_2$; or (2) according to a Gaussian measure on the bigger $\mathbb{R}^{\mathbb{N}}$, e.g. $W \sim \mathcal{N}(\mu, I)$ by which we mean the measure constructed as the product of a sequence $\mathcal{N}(\mu_i, 1)$ of independent real-valued Gaussians with unit variance. These two possibilities are mutually exclusive since the first choice gives a measure on $\mathbb{R}^{\mathbb{N}}$ whose mass is supported on $\ell_2$, while the second choice leads to a measure on $\mathbb{R}^{\mathbb{N}}$ supported outside of $\ell_2$. A good reference for this topic is Bogachev [1998].

Let us go with the second choice: $\mathcal{N}(0, I) = \bigotimes_{i=1}^{\infty} \mathcal{N}(0, 1)$, a 'standard Gaussian' on $\mathbb{R}^{\mathbb{N}}$. This is a legitimate probability measure on $\mathbb{R}^{\mathbb{N}}$ (by Kolmogorov's Extension theorem). But it is supported outside of $\ell_2$, so when sampling a $W \in \mathbb{R}^{\mathbb{N}}$ according to this measure, with probability one such $W$ will be outside of our feature space $\ell_2$. Then we have to wonder about the meaning of $\langle W, \cdot \rangle$ when $W$ is not in the Hilbert space carrying this inner product.

Let us write $W = (\xi_1, \xi_2, \ldots)$ a sequence of i.i.d. standard (real-valued) Gaussian random variables. Let $x = (x_1, x_2, \ldots) \in \ell_2$, and consider the formal expression $\langle x, W \rangle = \sum_{i=1}^{\infty} x_i \xi_i$. Notice that

$$\sum_{i=1}^{\infty} \mathbb{E}[|x_i \xi_i|^2] = \sum_{i=1}^{\infty} |x_i|^2 < \infty \,.$$

Then (see e.g. Bogachev [1998], Theorem 1.1.4) our formal object $\langle x, W \rangle = \sum_{i=1}^{\infty} x_i \xi_i$ is actually well-defined in the sense that the series is convergent almost surely (i.e. with probability one), although as we pointed out such $W$ is outside $\ell_2$.

## 5.1 Predicting with the Gaussian random classifier

Let $W_n = \mathrm{SVM}(S_n)$ be the weight vector found by running SVM on the sample $S_n$. We write it as $W_n = \sum_{i=1}^{n} \alpha_i Y_i \phi(X_i)$. Let $\kappa(\cdot, \cdot)$ be the kernel doing the "kernel trick."

Also as above let $W$ be a Gaussian random vector in $\mathbb{R}^{\mathbb{N}}$, and write it as $W = \sum_{j=1}^{\infty} \xi_j e_j$ with $\xi_1, \xi_2, \ldots$ i.i.d. standard Gaussians. As usual $e_j$ stands for the canonical unit vectors having a 1 in the $j$th coordinate and zeros elsewhere.

For an input $x \in \mathcal{X}$ with corresponding feature vector $\phi(x) \in \mathcal{H}$, we predict with

$$\langle W_n + W, \phi(x) \rangle = \sum_{i=1}^{n} \alpha_i Y_i \kappa(X_i, x) + \sum_{j=1}^{\infty} \xi_j \langle e_j, \phi(x) \rangle \,.$$

This is well-defined since

$$\sum_{i=1}^{\infty} \mathbb{E}[(\xi_j \langle e_j, \phi(x) \rangle)^2] = \sum_{i=1}^{\infty} (\langle e_j, \phi(x) \rangle)^2 = \|\phi(x)\|^2 \,,$$

so the series $\sum_{j=1}^{\infty} \xi_j \langle e_j, \phi(x) \rangle$ converges almost surely (Bogachev [1998], Theorem 1.1.4).

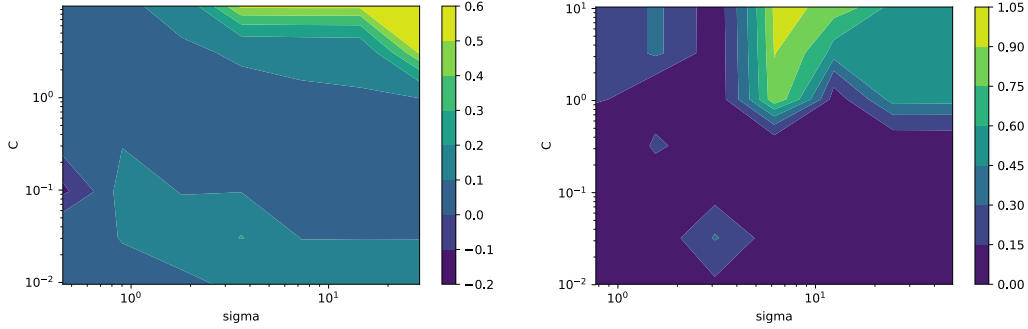

Figure 1: Tightness of P@O bound on PIM (left) and RIN (right) shown as the difference between the bound and the test error of the underlying randomized classifier. Smaller values are preferred.

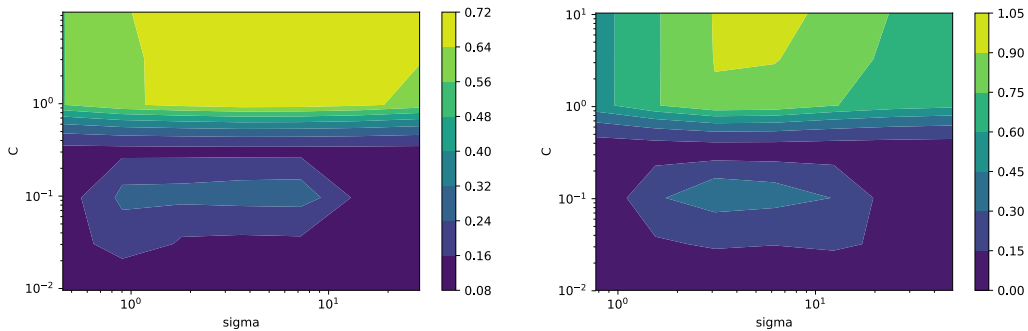

Figure 2: Tightness of P@EW bound (the bound derived here) on PIM (left) and RIN (right) shown as the difference between the bound and the test error of the underlying randomized classifier. Smaller values are preferred.

## 6 Experiments

The purpose of the experiments was to explore the strengths and potential weaknesses of our new bound in relation to the previous alternatives, as well as to explore the bound's ability to help model selection. For this, to facilitate comparisons, taking the setup of Parrado-Hernández et al. [2012], we experimented with the five UCI datasets described there. However, we present results for PIM and RIN only, as the results on the other datasets mostly followed the results on these and in a way these two datasets are the most extreme. In particular, they are the smallest and largest with dimensions $768 \times 8$ (768 examples, and 8 dimensional feature space), and $7200 \times 20$, respectively.

**Model and data preparation** We used an offset-free SVM classifier with a Gaussian RBF kernel $\kappa(x, y) = \exp(-\|x - y\|_2^2/(2\sigma^2))$ with RBF width parameter $\sigma > 0$. The SVM used the so-called standard SVM-C formulation which multiplies the total (hinge) loss by $C > 0$; the conversion to our formulation (svm) is given by $C = \frac{1}{\lambda n}$ where $n$ is the number of training examples and $\lambda > 0$ is the penalty factor. The datasets were split into a training and a test set using the `train_test_split` method of `scikit`, keeping $80\%$ of the data for training and $20\%$ for testing.

**Model parameters** Following the procedure suggested in Section 2.3.1 of Chapelle and Zien [2005], we set up a geometric $7 \times 7$ grid over the $(C, \sigma)$-parameter space where $C$ ranges between $2^{-8}C_0$ and $2^2 C_0$ and $\sigma$ ranges between $2^{-3}\sigma_0$ and $2^3\sigma_0$, where $\sigma_0$ is the median of the Euclidean distance between pairs of data points of the training set, and given $\sigma_0$, $C_0$ is obtained as the reciprocal value of the empirical variance of data in feature space underlying the RBF kernel with width $\sigma_0$. The grid size was selected for economy of computation. The grid lower and upper bounds for $\sigma$ were ad-hoc, though they were inspired by the literature, while for the same for $C$, we enlarged the lower range to focus on the region of the parameter space where the stability-based bounds have a better chance to be effective: In particular, the stability-based bounds grow with $C$ in a linear fashion, with a coefficient that was empirically observed to be close to one.

**Computations** For each of the $(C, \sigma)$ pairs on the said grid, we trained an SVM-model using a Python implementation of the SMO algorithm of Platt [1999], adjusted to SVMs with no offset (Steinwart and Christmann [2008] argue that "the offset term has neither a known theoretical nor an empirical advantage" for the Gaussian RBF kernel). We then calculated various bounds using the obtained model, as well as the corresponding test error rates (recall that the randomized classifiers' test error is different than the test error of the SVM model that uses no randomization). The bounds compared were the two mentioned hinge-loss based bounds: The bound by Liu et al. [2017] and that of Bousquet and Elisseeff [2002]. In addition we calculated the P@O and (our) P@EW bound. When these latter were calculated we optimized the randomization variance parameter $\sigma^2_{\text{noise}}$ by minimizing error estimate obtained from the respective bound (the KL divergence was inverted numerically). Further details of this can be found in Appendix D.

**Results** As explained earlier our primary interest is to explore the various bounds strengths and weaknesses. In particular, we are interested in their tightness, as well as their ability to support model selection. As the qualitative results were insensitive to the split, results for a single "random" (arbitrary) split are shown only.

*Tightness* The hinge loss based bounds gave trivial bounds over almost all pairs of $(C, \sigma)$. Upon investigating this we found that this is because the hinge loss takes much larger values than the training error rate unless $C$ takes large values (cf. Fig. 3 in Appendix D). However, for large values of $C$, both of the bounds are vacuous. In general, the stability based bounds (Liu et al. [2017], Bousquet and Elisseeff [2002] and our bound) are sensitive to large values of $C$. Fig. 1 show the difference between the P@O bound and the test error of the underlying respective randomized classifiers as a function of $(C, \sigma)$ while Fig. 2 shows the difference between the P@EW bound and the test error of the underlying randomized classifier. (Figs. 7 and 9 in the appendix show the test errors for these classifiers, while Figs. 6 and 8 shows the bound.) The meticulous reader may worry about that it appears that on the smaller dataset, PIM, the difference shown for P@O is sometimes negative. As it turns out this is due to the randomness of the test error: Once we add a confidence correction that accounts for the randomness of the small test set ($n_{\text{test}} = 154$) this difference disappears once we correct the test error for this. From the figures the most obvious difference between the bounds is that the P@EW bound is sensitive to the value of $C$ and it becomes loose for larger values of $C$. This is expected: As noted earlier, stability based bounds, which P@EW is an instance of, are sensitive to $C$. The P@O bound shows a weaker dependence on $C$ if any. In the appendix we show the advantage (or disadvantage) of the P@EW bound over the P@O bound on Fig. 10. From this figure we can see that on PIM, P@EW is to be preferred almost uniformly for small values of $C$ ($C \leq 0.5$), while on RIN, the advantage of P@EW is limited both for smaller values of $C$ and a certain range of the RBF width. Two comments are in order in connection to this: *(i)* We find it remarkable that a stability-based bound can be competitive with the P@O bound, which is known as one of the best bounds available. *(ii)* While comparing bounds is interesting for learning about their qualities, the bounds can be used together (e.g., at the price of an extra union bound).

*Model selection* To evaluate a bound's capability in helping model selection it is worth comparing the correlation between the bound and test error of the underlying classifiers. By comparing Figs. 6 and 7 with Figs. 8 and 9 it appears that perhaps the behavior of the P@EW bound (at least for small values of $C$) follows more closely the behavior of the corresponding test error surface. This is particularly visible on RIN, where the P@EW bound seems to be able to pick better values both for $C$ and $\sigma$, which lead to a much smaller test error (around 0.12) than what one can obtain by using the P@O bound.

# 7   Discussion

We have developed a PAC-Bayes bound for randomized classifiers. We proceeded by investigating the stability of the hypothesis learned by a Hilbert space valued algorithm. A special case being SVMs. We applied our main theorem to SVMs, leading to our P@EW bound, and we compared it to other stability-based bounds and to a previously known PAC-Bayes bound. The main finding is that perhaps P@EW is the first nontrivial bound that uses (uniform) hypothesis stability. Our work can be viewed as contributing to a line of research that aims to develop 'self-bounding algorithms' (Freund [1998], Langford and Blum [2003]) in the sense that besides producing a predictor the algorithm also creates a performance certificate based on the available data.

## Acknowledgements

Omar Rivasplata is sponsored by DeepMind via an Overseas Impact Studentship to undertake grad studies at UCL Department of Computer Science. Csaba Szepesvári gratefully acknowledges the Alberta machine intelligence institute (Amii), with funding from Alberta Innovates – Technology Futures and from the Natural Sciences and Engineering Research Council of Canada. Shiliang Sun is supported by the NSFC Project 61673179, and Shanghai Knowledge Service Platform Project ZF1213. John Shawe-Taylor acknowledges support of the UK Defence Science and Technology Laboratory (Dstl) and Engineering and Physical Sciences Research Council (EPSRC) under grant EP/R018693/1. This is part of the collaboration between US DOD, UK MOD and UK EPSRC under the Multidisciplinary University Research Initiative (MURI). This work was done in part while John Shawe-Taylor was visiting the Simons Institute for the Theory of Computing at UC Berkeley.

## Footnotes

[1] All spaces where random variables take values are assumed to be measurable spaces.

[2] $M_1(\mathcal{Z})$ denotes the set of all probability measures over the space $\mathcal{Z}$.

[3] Mathematicians write $\langle f, \nu \rangle \stackrel{\text{def}}{=} \int_{\mathcal{X} \times \mathcal{Y}} f(x, y) \, d\nu(x, y)$ for the integral of a function $f : \mathcal{X} \times \mathcal{Y} \to \mathbb{R}$ with respect to a (not necessarily probability) measure $\nu$ on $\mathcal{X} \times \mathcal{Y}$.

[4] Integrals with respect to $P_n$ evaluate as follows: $\int_{\mathcal{X} \times \mathcal{Y}} \ell(h(x), y) \, dP_n(x, y) = \frac{1}{n} \sum_{i=1}^{n} \ell(h(X_i), Y_i)$.

[5] For a list $\xi_1, \xi_2, \xi_3, \ldots$ and indexes $i < j$, we write $\xi_{i:j} = (\xi_i, \ldots, \xi_j)$, i.e., the segment from $\xi_i$ to $\xi_j$.

[6] Riesz representation theorem is behind this identification.

[7] $\mathcal{H}$ may be infinite-dimensional (e.g. Gaussian kernel).

[8] See Section 5 about the interpretation of Gaussian randomization for a Hilbert space valued algorithm.

[9]The Hilbert space norm, induced by the inner product of $\mathcal{H}$.

[10]Just to be sure: $\mathbb{R}^{\mathbb{N}}$ stands for the set of all infinite sequences of real numbers.

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
