[Supplementary Material]

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

# A  Proof of Lemma 6

Let $M_n = f(Z_1, \ldots, Z_n)$ be a function of the independent $\mathcal{Z}$-valued random variables $Z_1, \ldots, Z_n$, where the function $f : \mathcal{Z}^n \to \mathcal{H}$ maps into a separable Hilbert space $\mathcal{H}$. Let us write $M_n - \mathbb{E}[M_n]$ as the telescopic sum[11]

$$M_n - \mathbb{E}[M_n] = D_n + D_{n-1} + \cdots + D_1$$

where

$$D_i = \mathbb{E}[M_n | \mathcal{F}_i] - \mathbb{E}[M_n | \mathcal{F}_{i-1}]$$

and $\mathcal{F}_k = \sigma(Z_1, \ldots, Z_k)$ the $\sigma$-algebra generated by the first $k$ examples. Thus

$$\|M_n - \mathbb{E}[M_n]\|^2 = \sum_{i=1}^{n} \|D_i\|^2 + 2\sum_{i<j} \langle D_i, D_j \rangle \, .$$

We need $\mathbb{E}\|M_n - \mathbb{E}[M_n]\|^2$. Taking the expectation above makes the second sum disappear since for $i < j$ we have

$$\mathbb{E}[\langle D_i, D_j \rangle] = \mathbb{E}\big[\mathbb{E}[\langle D_i, D_j \rangle | \mathcal{F}_i]\big] = \mathbb{E}\big[\langle D_i, \mathbb{E}[D_j | \mathcal{F}_i] \rangle\big]$$

and clearly $\mathbb{E}[D_j | \mathcal{F}_i] = 0$ for $j > i$. Thus we have

$$\mathbb{E}\|M_n - \mathbb{E}[M_n]\|^2 = \mathbb{E}\sum_{i=1}^{n} \|D_i\|^2 \, . \tag{1}$$

Also recall the notation $\xi_{k:l} = (\xi_k, \ldots, \xi_l)$ for $k < l$. It will be used extensively in what follows.

Let us write the conditional expectations in terms of regular conditional probabilities:

$$\mathbb{E}[f(Z_{1:n}) | \mathcal{F}_i] = \int f(Z_{1:i}, z_{i+1:n}) \, dP_{Z_{i+1:n}|Z_{1:i}}(z_{i+1:n} | Z_{1:i}) \, .$$

The random variables are labelled with capitals. The lower case letters are for the variables of integration. We write $P_X$ for the distribution (probability law) of $X$.

Similarly

$$\mathbb{E}[f(Z_{1:n}) | \mathcal{F}_{i-1}] = \int f(Z_{1:i-1}, z_{i:n}) \, dP_{Z_{i:n}|Z_{1:i-1}}(z_{i:n} | Z_{1:i-1})$$

$$= \int f(Z_{1:i-1}, z_{i:n}) \, dP_{Z_i|Z_{1:i-1}}(z_i | Z_{1:i-1}) \cdot dP_{Z_{i+1:n}|Z_{1:i}}(z_{i+1:n} | Z_{1:i-1}, z_i) \, .$$

By independence, $P_{Z_{i+1:n}|Z_{1:i}} = P_{Z_{i+1:n}}$ and $P_{Z_i|Z_{1:i-1}} = P_{Z_i}$ (this latter is not really needed in the proof, but shortens the formulae). Hence,

$$D_i = \mathbb{E}[f(Z_{1:n}) | \mathcal{F}_i] - \mathbb{E}[f(Z_{1:n}) | \mathcal{F}_{i-1}] = \int f(Z_{1:i}, z_{i+1:n}) \, dP_{Z_{i+1:n}}(z_{i+1:n})$$

$$- \int f(Z_{1:i-1}, z_{i:n}) \, dP_{Z_i}(z_i) \, dP_{Z_{i+1:n}}(z_{i+1:n}) \, .$$

Then $D_i$ is equal to the integral w.r.t. $P_{Z_{i+1:n}}(z_{i+1:n})$ of

$$\int [f(Z_{1:i-1}, Z_i, z_{i+1:n}) - f(Z_{1:i-1}, z_i, z_{i+1:n})] \, dP_{Z_i}(z_i) \, .$$

Notice that in the integrand, only the $i$th argument differs. If $\|f(Z_{1:i-1}, Z_i, z_{i+1:n}) - f(Z_{1:i-1}, z_i, z_{i+1:n})\| \leq c$, then $\|D_i\| \leq c$. Thus bounded differences for $f(Z_{1:n})$ implies bounded martingale differences (in norm).

Finally, using Jensen's inequality and (1), and the bounded differences assumption:

$$\mathbb{E}\|M_n - \mathbb{E}[M_n]\| \leq \sqrt{\mathbb{E}\|M_n - \mathbb{E}[M_n]\|^2} \leq \sqrt{\sum_{i=1}^{n} c_i^2} \, .$$

# B  The average empirical error for Gaussian random linear classifiers

Let $Q = \mathcal{N}(w_0, I)$, a Gaussian with center $w_0 \in \mathbb{R}^d$ and covariance matrix the identity $d \times d$. The average empirical error is to be calculated as

$$R(Q, P_n) = \int_{\mathcal{X} \times \mathcal{Y}} \tilde{F}\left(\frac{y\, w_0^\top \phi(x)}{\|\phi(x)\|}\right)\, dP_n(x, y) \tag{2}$$

where $\tilde{F} = 1 - F$ and $F$ is the standard normal cumulative distribution

$$F(x) = \int_{-\infty}^{x} \frac{1}{\sqrt{2\pi}} e^{-u^2/2}\, du\,. \tag{3}$$

Recall that $P_n = \frac{1}{n} \sum_{i=1}^{n} \delta_{(X_i, Y_i)}$ is the empirical measure on $\mathcal{X} \times \mathcal{Y}$ associated to the $n$ training examples, and the integral with respect to $P_n$ evaluates as a normalized sum.

In this section we write the derivation of (2).

To make things more general let $Q = \mathcal{N}(w_0, \Sigma)$, a Gaussian with center $w_0 \in \mathbb{R}^d$ and covariance matrix $\Sigma$. We'll write $G_{(w_0, \Sigma)}$ for the corresponding Gaussian measure on $\mathbb{R}^d$. But to make notation simpler, lets work with input vectors $x$ (instead of feature vectors $\phi(x) \in \mathcal{H}$). This is in the context of binary classification, so the labels are $y \in \{\pm 1\}$. The classifier $c_w(\cdot) = \text{sign}(\langle w, \cdot \rangle)$ is identified with the weight vector $w$. The loss on example $(x, y)$ can be written as

$$\ell(c_w(x), y) = \mathbf{1}(c_w(x) \neq y) = \frac{1 - \text{sign}(y\langle w, x\rangle)}{2}\,.$$

We'll talk about the empirical error of $w$, namely $R(w, P_n) = \int_{\mathcal{X} \times \mathcal{Y}} \mathbf{1}(c(x) \neq y)\, dP_n(x, y)$. The average empirical error when choosing a random weight $W$ according to $Q$ is:

$$R(Q, P_n) = \int_{\mathbb{R}^d} R(w, P_n)\, dG_{(w_0, \Sigma)}(w)\,.$$

Plugging in the definition of $R(w, P_n)$ and swapping the order of the integrals and using the above formula for the loss, the right hand side is

$$\int_{\mathbb{R}^d} \int_{\mathcal{X} \times \mathcal{Y}} \mathbf{1}(c(x) \neq y)\, dP_n(x, y)\, dG_{(w_0, \Sigma)}(w)$$

$$= \int_{\mathcal{X} \times \mathcal{Y}} \int_{\mathbb{R}^d} \mathbf{1}(c(x) \neq y)\, dG_{(w_0, \Sigma)}(w)\, dP_n(x, y)$$

$$= \int_{\mathcal{X} \times \mathcal{Y}} \int_{\mathbb{R}^d} \frac{1 - \text{sign}(y\langle w, x\rangle)}{2}\, dG_{(w_0, \Sigma)}(w)\, dP_n(x, y)$$

$$= \int_{\mathcal{X} \times \mathcal{Y}} \frac{1}{2}\left(1 - A(x, y)\right)\, dP_n(x, y)$$

where for a fixed pair $(x, y)$ we are writing

$$A(x, y) = \int_{\mathbb{R}^d} \text{sign}(y\langle w, x\rangle)\, dG_{(w_0, \Sigma)}(w)\,.$$

Decompose into two terms:

$$A(x, y) = \int_{y\langle w, x\rangle > 0} dG_{(w_0, \Sigma)}(w) - \int_{y\langle w, x\rangle < 0} dG_{(w_0, \Sigma)}(w)$$

and notice that for the random vector $W \sim \mathcal{N}(w_0, \Sigma)$ we have $\mathbb{E}[y\langle W, x\rangle] = y\langle w_0, x\rangle$ and $\mathbb{E}[(y\langle W, x\rangle)^2] = \|x\|_{\Sigma}^2 + (\langle w_0, x\rangle)^2$, so the functional $y\langle W, x\rangle$ has a 1-dimensional Gaussian dis-

tribution with mean $y\langle w_0, x\rangle$ and variance $\|x\|_\Sigma^2 = \langle \Sigma x, x\rangle$. Then

$$\int_{y\langle w,x\rangle>0} dG_{(w_0,\Sigma)}(w) = \Pr[y\langle W, x\rangle > 0]$$

$$= \Pr\left[\frac{y\langle W,x\rangle - y\langle w_0,x\rangle}{\|x\|_\Sigma} > \frac{-y\langle w_0,x\rangle}{\|x\|_\Sigma}\right]$$

$$= \Pr\left[\mathcal{N}(0,1) > \frac{-y\langle w_0,x\rangle}{\|x\|_\Sigma}\right]$$

$$= \Pr\left[\mathcal{N}(0,1) < \frac{y\langle w_0,x\rangle}{\|x\|_\Sigma}\right] = F\left(\frac{y\langle w_0,x\rangle}{\|x\|_\Sigma}\right).$$

Then

$$A(x,y) = 2F\left(\frac{y\langle w_0,x\rangle}{\|x\|_\Sigma}\right) - 1$$

and

$$1 - A(x,y) = 2 - 2F\left(\frac{y\langle w_0,x\rangle}{\|x\|_\Sigma}\right) = 2\tilde{F}\left(\frac{y\langle w_0,x\rangle}{\|x\|_\Sigma}\right).$$

Altogether this gives

$$R(Q, P_n) = \int_{\mathcal{X}\times\mathcal{Y}} \tilde{F}\left(\frac{y\langle w_0,x\rangle}{\|x\|_\Sigma}\right) dP_n(x,y).$$

Notice that using $\Sigma = I$ (identity) and $\phi(x)$ instead of $x$ this gives (2).

REMARK: Langford [2005] uses a $Q$ which is $\mathcal{N}(\mu, 1)$ along the direction of a vector $w$, and $\mathcal{N}(0,1)$ in all directions perpendicular to $w$. Such $Q$ is a Gaussian centered at $w_0 = \mu w/\|w\|$, giving his formula

$$R(Q, P_n) = \int_{\mathcal{X}\times\mathcal{Y}} \tilde{F}\left(\mu\frac{y\, w^\top \phi(x)}{\|w\|\,\|\phi(x)\|}\right) dP_n(x,y).$$

## C  SVM weight vector: clarification about formulations

We have a sample of size $n$. In the standard implementation the weight vector $W_n(C)$ found by SVM is a solution of the following optimization problem:

$$W_n(C) = \arg\min_w \left(\frac{1}{2}\|w\|_\mathcal{H}^2 + C\sum_{i=1}^n \xi_i\right). \tag{svm1}$$

In our paper the weight vector $W_n^{\mathrm{OURS}}(\lambda)$ found by SVM is a solution of the following optimization problem:

$$W_n^{\mathrm{OURS}}(\lambda) = \arg\min_w \left(\frac{\lambda}{2}\|w\|_\mathcal{H}^2 + \frac{1}{n}\sum_{i=1}^n \xi_i\right). \tag{svm2}$$

In Bousquet and Elisseeff [2002] and Liu et al. [2017] the weight vector $W_n^{\mathrm{B\&E}}(\lambda)$ found by SVM is a solution of the following optimization problem:

$$W_n^{\mathrm{B\&E}}(\lambda) = \arg\min_w \left(\lambda\|w\|_\mathcal{H}^2 + \frac{1}{n}\sum_{i=1}^n \xi_i\right). \tag{svm3}$$

The minimum is over $w \in \mathcal{H}$ (an appropriate Hilbert space) and subject to some constrains for the $\xi_i$'s in all cases. The relation between them is:

- $W_n^{\mathrm{OURS}}(\lambda) = W_n^{\mathrm{B\&E}}(\lambda/2)$

- $W_n^{\mathrm{B\&E}}(\lambda) = W_n(C)$ with $C = \frac{1}{2n\lambda}$

- $W_n^{\mathrm{OURS}}(\lambda) = W_n(C)$ with $C = \frac{1}{n\lambda}$

# D  Details for experiments

In this section we show further details and results that did not fit the main body of the paper.

## D.1  Details of optimizing $\sigma^2_{\text{noise}}$

This optimization is "free" for the P@O bound as the bound is uniform over $\sigma^2_{\text{noise}}$. In the P@EW bound we adjusted the failure probability $\delta$ to accommodate the multiple evaluations during the optimization by replacing it with $\delta/(\tau(\tau+1))$, where $\tau$ is the number of times the P@EW bound is evaluated by the optimization procedure. A standard union bound argument shows that the adjustment to $\delta$ makes the resulting bound hold with probability $1 - \delta$ regardless the value of $\tau$. The SLSQP method implemented in SCIPY was used as an optimizer, with an extra outer loop that searched for a suitable initialization, as SLSQP is a gradient based method and the P@O bound can be quite "flat". The same problem did not appear for the P@EW bound. The attentive reader may be concerned that if $\tau$ gets large values, we, in a way, are optimizing the "wrong bound". To check whether this is a possibility, we also computed the "union bound penalty" for decreasing $\delta$ by the factor $\tau(\tau+1)$ as the difference between the (invalid) bound where $\delta$ is unchanged and the bound where $\delta$ is decreased and found that the penalty was generally orders of magnitudes smaller than the risk estimate. Nevertheless, this may be a problem when the risk to be estimated is very small, which we think is not very common in practice.

## D.2  Additional figures

Figure 3: Hinge loss on PIM (left) and RIN (right). For large values of $C$, the hinge loss is reasonable, but this is not the case for small values.

Figure 4: The bound of Liu et al. [2017] on PIM (left) and RIN (right). The bound is almost always vacuous for reasons described in the text.

Figure 5: The bound of Bousquet and Elisseeff [2002] on PIM (left) and RIN (right). The bound is almost always vacuous for reasons described in the text.

Figure 6: The P@O bound on PIM (left) and RIN (right).

Figure 7: Test error of the randomized classifiers underlying the P@O bound on PIM (left) and RIN (right).

Figure 8: The P@EW bound on PIM (left) and RIN (right).

Figure 9: Test error of the randomized classifiers underlying the P@EW bound on PIM (left) and RIN (right).

Figure 10: Advantage of the P@EW bound to the P@O bound on PIM (left) and RIN (right): The figure shows the difference between the P@O bound and the P@EW bound. Where this is positive, P@EW is to be preferred, while where it is negative, P@O is to be preferred.