[Reviews · NeurIPS 2018]

Reviewer 1



Summary: This paper proposes a theoretical analysis of stable algorithms by means of the PAC-Bayesian theory for randomized classifiers. The originality come from the fact that the authors use a stability with respect to a distance (norm) between hypothesis instead of the classic stability with respect to loss functions. The authors propose a new PAC-Bayes bound in the context of Gaussian randomization and exhibits a special case for SVM classifiers. The bound is discussed and compared with other stability based bounds. Pros/Cons Pros: -New PAC-Bayes bound that appears to be informative and tight -In my point of view, the paper can offer a basis to other developments -As a side contribution, an extension of the McDiarmid inequality to vector-valued functions is proposed. Cons: -A more precise comparison/discussion to related notion of stability should be made -A detailed discussion with other PAC-Bayesian bounds is missing -Results specialized to Gaussian randomization, perspectives of development are not well developed Overall: Comments +The notion of hypothesis sensitivity coefficients (Definition 1) seem very similar to the notion of uniform argument stability of [Liu et al., 2017] (Definition 2). Could the author discuss the difference between the two formulations. There is also the notion of collective stability which is applied to vector-valued functions: Ben London et al.,PAC-Bayesian Collective Stability. AISTATS 2014 The authors should also discuss this notion and the point is also related to the comparison of PAC-Bayes results (see below) +One appealing aspect of algorithmic stability is to provide some guarantees for the algorithm used to learn the hypothesis, and more technically with respect to the loss used. By using the hypothesis sensitivity coefficients, you are not directly dependent on the algorithm and the loss used but it seems to me that there is an indirect dependency. It could be interesting to discuss this dependency. It seems natural to think that if we have the hypothesis sensitivity then we have an hypothesis stability for the corresponding loss but what about the other side. Considering randomized algorithms, this dependency (if confirmed) seems to influence the mean hypothesis of the distribution. Can we say that the learning algorithm can be interpreted as a way to bias the distribution. +The authors have compared their work with previous bounds using a notion of stability which is good and interesting. But I am a bit disappointed that there is no real discussion with PAC-Bayesian bounds and in particular those using Gaussian distributions (mentioned for example in the paper of Germain, Lacasse, Laviolette, Marchand, Roy; JMLR 2015) It appears that the result seem to have a better convergence than existing bounds. In general, for having a better rate than O(1/\sqrt(n)) one must rely on some (strong) assumption(s). Since the uniform stability framework and the classic PAC-Bayesian theorems (as used in the paper) with Gaussian distributions do not allow to obtain better rates easily taken separately, I wonder what is the assumption setting that makes this new result possible. Is is the particular combination of stability over coefficients with the assumption of Gaussian distributions over them or is there another underlying assumption? +This is maybe related to the question above, but how the capacity to define good Gaussian distributions over the Hilbert space of classifiers (as discussed in Appendix E) is key for the result and in particular for developing interesting perspectives? +The result is based on a Seeger/Langford formulation, is it possible to obtain results based on other PAC-Bayesian theorems such as McAllester1999 or Catoni2007 for example? From new theorems, will it be possible to develop new learning algorithms specifically adapted to the PAC-Bayesian setting ? +The bound found by the authors is penalized by small variances. This point is interesting because it illustrates that for a randomized algorithm (which can also be interpreted as a majority vote) the classifiers involved must have some diversity. If you have a large variance, the bound will become better and the average hypothesis will have less importance. Do the authors that the presence of the \sigma^2 parameter is reasonable and if there is room for improvement or a better understanding of randomized/majority vote algorithms? The notion of diversity appears in the C-bound of (Lacasse et al.,NIPS'07), I do not know if some relationships can be made here. +line 44 notation M_1(.) could be introduced at this line. +Note that the reference [Liu et al.,2017] has been published at ICML 2017. #### After Rebuttal #### Thank you for your answers. I think that the authors should precise the connections/differences between the works of Liu et al. and London et al. (at least in supplementary for example). The discussion with Germain et al,09 is also necessary. For the sake of completeness, I would suggest to the authors to add the proof of the expression of the Kullback-Leibler divergence for Gaussians over Hilbert spaces, line 174 (for example in supplementary). The discussion about perspectives were rather short in the paper, the authors may try to improve the discussion about possible perspectives.

Reviewer 2



Summary: The paper presents a new PAC-Bayes bound for stable Hilbert space valued algorithms that is obtained by using a specifically defined prior distribution. In case of SVM, the proven bound is compared to the respective results from the literature. In addition to discussing the formal comparison, the authors show the empirical behavior of the new bound on two datasets from UCI. Quality: I like the idea of using a specifically designed prior distribution in PAC-Bayes framework to show a better bound. The proofs appear correct to me except for one place: I am not sure that the formula for the KL-divergence between the Guassian remains the same in the formula without a number between page 174 and 175 (it is also not clear what option for defining the Guassian distribution from supplement E is used) and I would like the authors to address that in their response. Moreover, I don't like that all quite important technical details regarding the definition of the Gaussian distribution over the Hilbert space are moved to the supplement and I think these details needs to be a part of the main paper. Clarity: In general the paper is well-written. The exposition is clear except for the part when the authors switch to discussing distributions over the Hilbert spaces. Some relevant work is not discussed though, e.g. London, "Generalization Bounds for Randomized Learning with Application to Stochastic Gradient Descent" and other work on the mix of stability with PAC-Bayes. Originality: The presented results are new to my best knowledge. Significance: Given that the authors address the issue of Guassians over Hilbert spaces, I think the paper presents an interesting approach for mixing PAC-Bayes with stability arguments and should have a positive impact.

Reviewer 3



**Summary and main remarks** The manuscript investigates binary classification and proposes PAC-Bayesian bounds relying on a notion of stability. Stability has attracted a lot of efforts in the past two decades and loosely can be interpreted as "when the input training data changes by a small amount, the resulting algorithm should also change at most by a small amount". The notion used in the manuscript (Definition 1, lines 79--81) is of similar flavor as to what is known as hypothesis stability in the literature (see Bousquet and Elisseeff, 2002), but rather on the learned hypotheses. The authors call an algorithm $A$ $\beta_m$-stable iff the sup over index $i$ of the difference between the algorithm trained on an $m$ sample and the algorithm trained on the exact same $m$ sample but where data point $i$ is an independent copy, is bounded by $\beta_m$ in the sense of the norm induced by the Hilbert structure. This is the strongest notion of stability. As it is well-known in the literature and also noted by the authors (this is the exact title of subsection 4.3), stability implies concentration. Therefore, requiring that an algorithm $A$ is $\beta_m$-stable straightforwardly yields a concentration result on its empirical risk towards its risk. Such a concentration result is a cornerstone of PAC-Bayes and the authors derive a PAC-Bayesian bound (Theorem 2, page 3) which upper-bounds the discretized Kullback-Leibler (KL) divergence between the empirical risk of a randomized classifier and its risk, by a term of magnitude $\beta_m^2 / \sigma^2+\log(m)/m$. This bound is vacuous whenever $\beta_m$ does not decay as $m$ grows -- the authors remark that for SVM, Bousquet and Elisseeff proved in 2002 that $\beta_m$ is upper-bounded by $1/n$, and the bound in Theorem 2 becomes non-vacuous with a rate of order $\log(m)/m$. I feel the manuscript might be lacking a discussion on which other algorithm satisfies this regime. A single example (SVM) might not be convincing enough. The manuscript then compares its bound to three others in the literature, and discusses at lenghth its merits. I have found this discussion very interesting. The manuscript closes with the proofs of claimed results and a small numerical experiments section. I have checked the proofs and as far as I can tell they appear to have no flaw. **Overall assessment** The paper is well-written and contains interesting contributions. Past years have seen a resurgence of PAC-Bayesian bounds, and stability is a common framework for deriving generalization bounds; however it seems that the manuscript is the first to combine both approaches. My main concern is that the scope of the manuscript might be limited, as only one example of learning algorithm which complies with the stability notion introduced is given (SVM). **Specific remarks** - typo: repeated "that" line 33 - Some improper capitalization of words such as Bayes, PAC in the references. - Lines 292--293: I agree that the bound proposed in the manuscript seems to be the first PAC-Bayesian bound relying on stability. However it seems that Celisse and Guedj (2016) derive PAC (not Bayes) generalization bounds using a notion of stability ($L^q$) which is weaker than the notion used in the manuscript. I would suggest to rephrase the last sentence (and other such claims elsewhere in the manuscript) to avoid confusion, and explicitly state that the manuscript contains "the first nontrivial PAC-Bayesian bound that uses stability". [Rebuttal acknowledged.]